# Generating dual structurally and functionally skin-mimicking hydrogels by crosslinking cell-membrane compartments

Feng Wu [1,2,5], Yusheng Ren[1,2,5], Wenyan Lv[2,3], Xiaobing Liu[2,3], Xinyue Wang[2], Chuhan Wang[4], Zhenping Cao[2], Jinyao Liu [2] ✉, Jie Wei [1] ✉ & Yan Pang [4] ✉

The skin is intrinsically a cell-membrane-compartmentalized hydrogel with high mechanical strength, potent antimicrobial ability, and robust immunological competence, which provide multiple protective effects to the body. Methods capable of preparing hydrogels that can simultaneously mimic the structure and function of the skin are highly desirable but have been proven to be a challenge. Here, dual structurally and functionally skin-mimicking hydrogels are generated by crosslinking cell-membrane compartments. The crosslinked network is formed via free radical polymerization using olefinic double bond-functionalized extracellular vesicles as a crosslinker. Due to the dissipation of stretching energy mediated by vesicular deformation, the obtained compartment-crosslinked network shows enhanced mechanical strength compared to hydrogels crosslinked by regular divinyl monomers. Biomimetic hydrogels also exhibit specific antibacterial activity and adequate ability to promote the maturation and activation of dendritic cells given the existence of numerous extracellular vesicle-associated bioactive substances. In addition, the versatility of this approach to tune both the structure and function of the resulting hydrogels is demonstrated through introducing a second network by catalyst-free click reaction-mediated crosslinking between alkyne-double-ended polymers and azido-decorated extracellular vesicles. This study provides a platform to develop dual structure- and function-controllable skin-inspired biomaterials.

As the largest human organ with high mechanical strength, potent antimicrobial ability, and robust immunological competence, the skin directly interfaces with external environments and performs as an important physical and immunological barrier to protect the body from various invasions[1–3]. Methods capable of mimicking the structure or function of the skin are highly desirable to prepare appealing materials that can be applied in a wide range of applications from tissue regeneration[4,5], wearable devices[6], soft robotics[7], health monitoring[8] to intelligent medical diagnosis[9]. To mimic the function of the skin, the vast majority of studies are based on hybridization, mainly by introducing diversified inorganic[10] or conductive[11,12] nanomaterials to different gel materials, such as hydrogels[13–15], organic gels[16], and

[1]Shanghai Key Laboratory of Advanced Polymeric Materials, School of Materials Science and Engineering, East China University of Science and Technology, Shanghai, China. [2]State Key Laboratory of Systems Medicine for Cancer, Shanghai Cancer Institute, Shanghai Key Laboratory for Nucleic Acid Chemistry and Nanomedicine, Institute of Molecular Medicine, Renji Hospital, School of Medicine, Shanghai Jiao Tong University, Shanghai, China. [3]College of Chemistry and Materials Science, Shanghai Normal University, Shanghai, China. [4]Shanghai Key Laboratory of Orbital Diseases and Ocular Oncology, Department of Ophthalmology, Shanghai Ninth People's Hospital, School of Medicine, Shanghai Jiao Tong University, Shanghai, China. [5]These authors contributed equally: Feng Wu, Yusheng Ren. ✉e-mail: jyliu@sjtu.edu.cn; jiewei7860@sina.com; yanpang@sjtu.edu.cn

ionic liquid gels[17]. To date, numerous skin-mimetic materials with attractive functions, such as multimodal sensing capability[18–20], sweating ability[21], enhanced mechanical strength[22], ultrafast self-healing[23,24], controllable chromotropic capability[25,26], and steady bio-adhesion[27], have been developed with the help of this strategy. To simulate the structure of the skin, previous studies majorly rely on the integration of multiple components or layers with different features into a coherent system[28,29]. For example, to acquire specific cutaneous functions, several modules including gradient pore skeleton[18,30], miniaturized artificial devices[31], and nerve-like nanonetwork[32] have been integrated into the matrix materials to mimic the sweat glands and sensory receptors in the skin. Despite these elegant achievements, approaches that are able to mimic the structure of the skin to prepare bioinspired materials with skin-like functions, particularly with advanced mechanical and biological properties, have been rarely reported yet.

The skin comprises a three-layered cell-based structure containing the epidermis, dermis, and hypodermis[33], which are composed of keratinocytes, fibroblasts, and adipocytes[34,35], respectively. These cells are embedded within an extracellular matrix (ECM) consisting of multilamellar lipids, collagen, elastin, and/or hyaluronic acid[36]. Essentially, the skin can be simplified as a composite hydrogel material, in which a large number of cells are connected by the ECM. The multiple interactions between the isolated cell-membrane compartments and their surrounding ECM endow the formed network with a favorable mechanical strength. Meanwhile, owing to the existence of living cells that can release diverse bioactive substances[37], this composite hydrogel material possesses special biological characteristics, such as antimicrobial and immunological activities, forming a protective barrier to prevent against external invasions[38]. Therefore, synthetic strategies capable to construct cell-membrane-based networks are highly fascinating considering their potential to prepare hydrogel materials that can simultaneously mimic the structure and function of the skin.

Here, we describe the crosslinking of cell-membrane compartments to develop dual structurally and functionally skin-mimicking hydrogels (SFSHs) (Fig. 1). The crosslinked cell-membrane-based network is generated by free radical aqueous polymerization of acrylamide using olefinic double bond-decorated extracellular vesicles as a crosslinker, which is constructed through hydrophobic/hydrophilic interaction-mediated insertion of distearoylphosphatidylethanolamine-polyethylene glycol-acrylamide (DSPE-PEG-AM) into the vesicular cell membrane. In contrast to typical methylene diacrylamide-crosslinked polyacrylamide hydrogels, the obtained network exhibits largely increased mechanical strength due to the attachment of multiple polymer chains to a mechano-deformable compartment crosslinker, namely the double bond-functionalized vesicle, which can dissipate stretching energy via vesicular deformation. Since extracellular vesicles inherent numerous bioactive substances from their parent cells, the resulting SFSHs also show specific antibacterial effect and potent capability to provoke the maturation and activation of dendritic cells. We further demonstrate the versatility of this method to control both the structure and function of SFSHs via incorporating a second network by catalyst-free click reaction-enabled crosslinking between alkyne-double-ended PEG and azido-modified extracellular vesicles. Given the flexibility of this methodology, we anticipate that such a biomimetic paradigm can pave an avenue to prepare dual structure- and function-tunable skin-inspired biomaterials and beyond.

## Results and discussion

### Design and preparation of cell-membrane-compartmentalized network

We chose polyacrylamide as the matrix given its wide application as ECM analogue to develop skin-inspired materials as well as its biocompatibility and facile preparation[39,40]. On the other side, we selected extracellular vesicles as cell-membrane compartments due to the presence of both lipid bilayer structure and substantial bioactive substances inherited from parent cells[41,42]. To mimic the structure of the skin, extracellular vesicles were used as a multivalent crosslinker to crosslink the polyacrylamide chains, forming a cell-membrane-compartmentalized network. Considering the easy availability, *Escherichia coli* Nissle 1917 cell-derived outer membrane vesicles (OMVs) were employed as a model extracellular vesicle to fabricate SFSHs. OMVs were extracted and purified from the culture medium by ultracentrifugation. As shown in Supplementary Fig. 1A and B, the collected OMVs showed a spherical structure with a size distribution centered around 180 nm. In the following studies, the concentrate of OMVs with different particle numbers was quantified by nanoparticle tracking assay (NTA) measurement and the corresponding total protein amount was determined using bicinchoninic acid (BCA) protein quantification kit (Supplementary Fig. 1C, D). Typically, $1.44 \times 10^{12}$ OMV particles/ml contained a corresponding protein concentration of 2 mg/ml.

To play a crosslinker role, the purified vesicles were decorated with olefinic double bonds on the surface via supramolecular hydrophobic/hydrophilic interaction between DSPE-PEG-AM and the lipid bilayer of OMVs. The decoration was conducted at 37 °C, as which could disturb phospholipid arrangement to facilitate the insertion of DSPE molecules into the bilayer of OMVs[43]. A thiol-ene addition reaction mediated by sulfhydrated Cyanine5 (SH-Cy5) fluorescent dye was utilized to verify the resulting acrylamide-functionalized OMVs (termed as OMV-AM). As analyzed by flow cytometry (FCM), the fluorescence peak of OMV-AM significantly shifted to the higher intensity with approximately 3-fold elevated mean fluorescent intensity (MFI) after surface functionalization of alkenyl double bonds (Fig. 2A, B). Laser scanning confocal microscopy (LSCM) visualized that OMVs had no fluorescence originally, while the modified vesicles emitted prominent fluorescence (Fig. 2C). Both FCM and LSCM measurements suggested the successful preparation of alkenyl double bond-functionalized OMVs. Transmission electron microscope (TEM) imaging of OMV-AM displayed undetectable changes in the morphology of the spherical structure (Fig. 2D). After modification, the average particle size and zeta potential of OMV-AM increased from 187.9 to 230.7 nm and −16.4 to −9.5 mV, respectively (Fig. 2E, F). Compared to DLS measurement, the slightly decreased size observed by TEM could be attributed to the dehydration of OMVs during sample preparation. The increment in zeta potential might be ascribed to the introduction of DSPE-PEG-AM, which inserted the hydrophobic DSPE tails into the lipid bilayer of OMVs and exposed the hydrophilic PEG-AM heads containing positively charged secondary amine groups on the surface of OMVs.

SFSHs were prepared via free radical aqueous polymerization using acrylamide, OMV-AM, ammonium persulfate (APS), and *N*, *N*, *N'*, *N'*-tetramethylethylenediamine (TEMED) as monomer, crosslinker, initiator, and catalyst, respectively (Fig. 2G). Polyacrylamide hydrogels that were simply crosslinked by regular small molecular crosslinker of *N*, *N'*-methylenebisacrylamide (MBAA) without the addition of OMVs were used as controls. The microstructure and morphology of SFSHs were observed by scanning electron microscope (SEM). As observed in Fig. 2H and Supplementary Fig. 2, both SFSHs and the control polyacrylamide hydrogel after lyophilization exhibited a porous 3-dimentional (3D) structure under a low magnification. With the increase of magnification, these two hydrogels presented clear differences in structural morphology, in which the control hydrogel showed a layered structure along the thickness direction (Fig. 2I), while SFSHs exhibited numerous nanospheres scattered in the cross section (Fig. 2J). Upon the concentration of OMV-AM increasing from $1 \times 10^9$ to $1 \times 10^{12}$ particles/ml, SEM observation showed that the cross section was embedded with a large number of vesicles (Fig. 2K), highlighting the formation of the vesicle-compartmentalized network structure. As

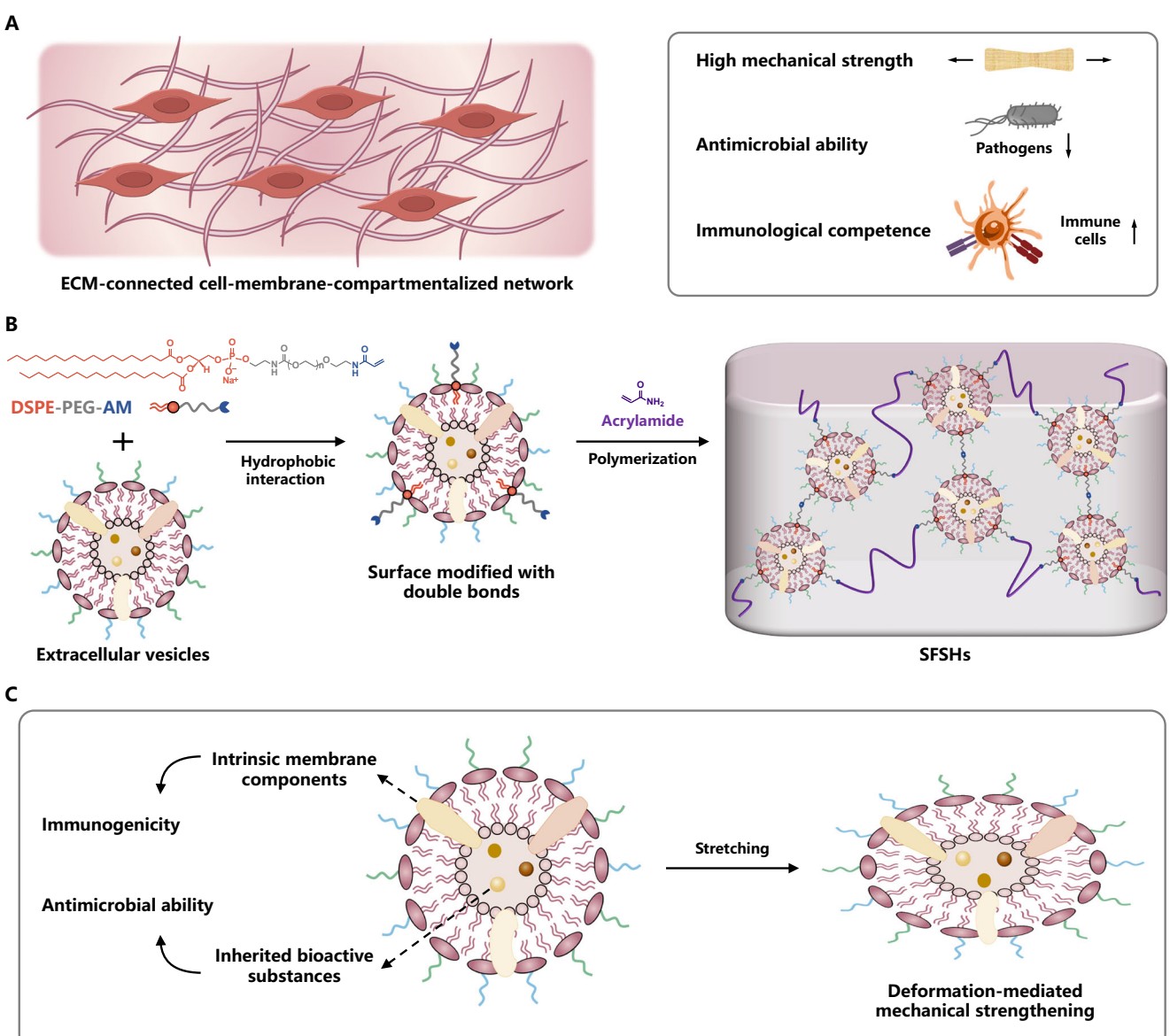

**Fig. 1 | Schematic illustration. A** Illustration of the skin, which is intrinsically an ECM-connected cell-membrane-compartmentalized hydrogel with high mechanical strength, potent antimicrobial ability, and robust immunological competence.

**B** Synthetic route of SFSHs. **C** Characteristics of SFSHs mediated by crosslinking extracellular vesicles.

visualized by 3D LSCM imaging, dense OMVs labeled with *N*-hydroxysuccinimide (NHS) ester-functionalized Cy5 fluorescent dye (Cy5-NHS) were dispersed uniformly inside the network (Fig. 2L), which verified that SFSHs with a biomimetic cell-compartmentalized structure to the skin were formed successfully.

### Enhanced mechanical strength of SFSHs

We systemically explored the influence of preparation conditions on the mechanical properties of SFSHs. Considering the critical role of crosslinker OMV-AM toward the mechanical performance of the network, we first investigated the reaction conditions including temperature, time, and operation procedures for the modification of OMVs. Interestingly, under the same experimental conditions, SFSHs basing on OMV-AM that was prepared by incubation OMVs with DSPE-PEG-AM at 37 °C rather than 25 °C and 50 °C presented the optimal tensile properties including the highest tensile stress, modulus, and strain (Fig. 3A–D). Note that these mechanical parameters of SFSHs decreased with incubation time of OMVs prolonging from 0.5 h to 1 h at 37 °C (Fig. 3E–H). It was also found that treatment with ultrasound

during the incubation could reduce the tensile properties of SFSHs. Namely, OMV-AM obtained after 0.5 h incubation with DSPE-PEG-AM at 37 °C without the treatment of ultrasound could be used to prepare SFSHs with the best toughness. We speculated that these conditions offered a moderate perturbation of the lipid bilayer of OMVs, which could promote the insertion of DSPE-PEG-AM into the vesicular cell membrane driven by supramolecular hydrophobic/hydrophilic interaction. As we known, the crosslinking density of a polymer network determines the mechanical properties of the hydrogel. That means the mechanical strength of SFSHs largely depends on the number of involved double bonds. We therefore studied the influences of both particle number of vesicles and the density of double bonds decorated on OMVs to optimize the mechanical performance of SFSHs. As indicated in Fig. 3I–L, the tensile stress and strain improved with OMV-AM concentration increasing from $1 \times 10^9$ to $1 \times 10^{12}$ particles/ml with the rest experimental conditions remained unchanged. Furthermore, under the same number of added vesicles, the tensile modulus of SFSHs increased with the solid content of DSPE-PEG-AM increasing from 0.8% to 8%, while the tensile strain decreased with the increase of

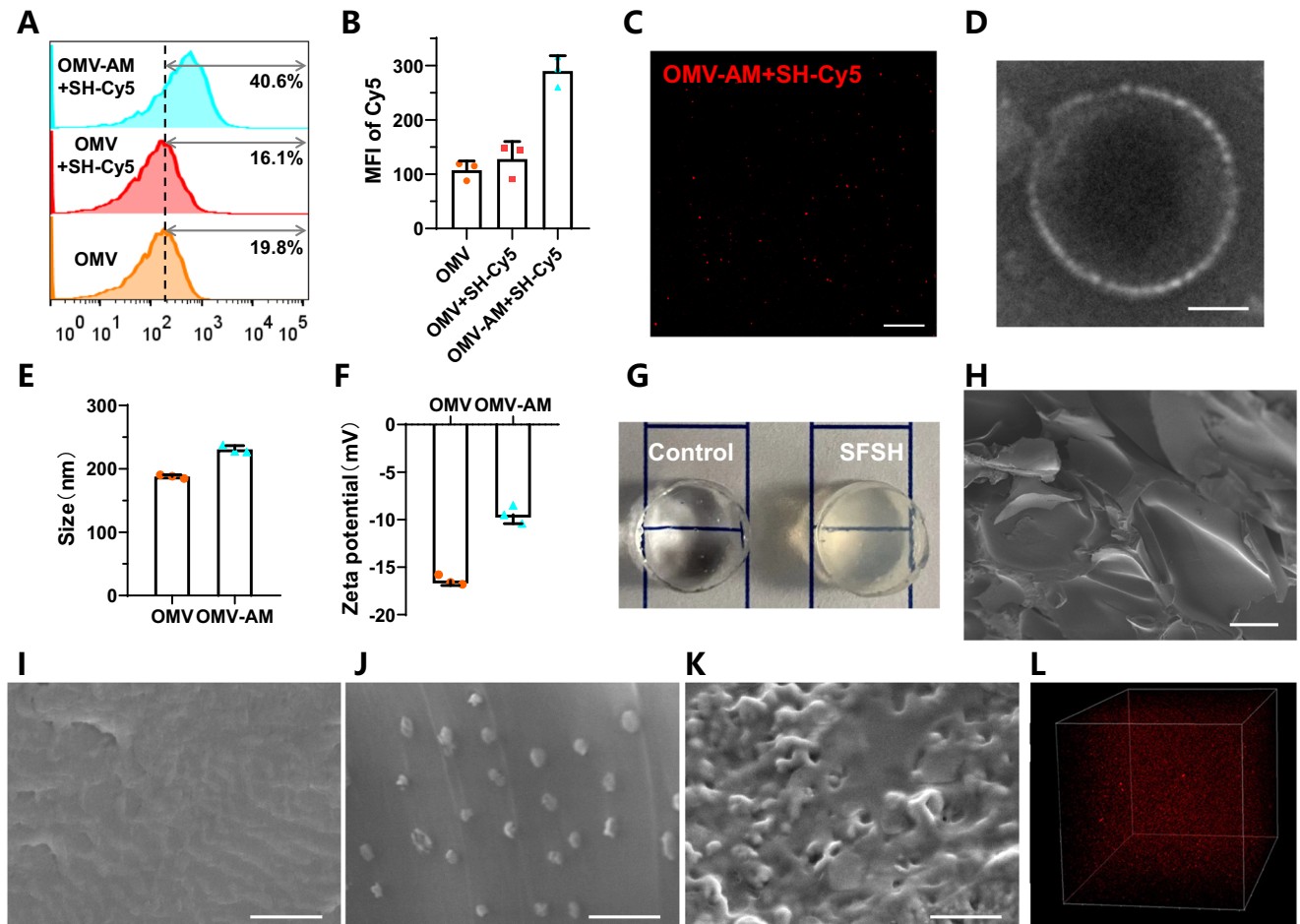

**Fig. 2 | Characterization of OMV-AM and SFSHs. A** FCM histograms of OMVs and OMV-AM after incubation with SH-Cy5 at 37 °C for 3 h, respectively. **B** MFI of Cy5-labeled OMVs and OMV-AM measured by FCM analysis. **C** A representative LSCM image of Cy5-labeled OMV-AM. Scale bar: 25 μm. **D** A typical TEM image of OMV-AM. Scale bar: 50 nm. **E** Average size and **F** zeta potential of OMVs and OMV-AM. **G** Digital photos of control polyacrylamide hydrogel and SFSHs. **H** A typical SEM image of lyophilized SFSHs. Magnification: × 90. Scale bar: 100 μm. SEM images of lyophilized **I** control polyacrylamide hydrogel and SFSHs with **J** 1 × 10⁹ and **K** 1 × 10¹² particles/ml of OMV-AM. Magnification: × 10000. Scale bar: 1 μm. **L** A representative 3D LSCM image of SFSHs with Cy5-labeled OMVs. Data are presented as mean values ± SD ($n = 3$, from independent experiments).

double bond density on OMV-AM (Fig. 3M–P). As screened, at a combination of OMV-AM concentration of $1.5 \times 10^{12}$ particles/ml and DSPE-PEG-AM solid content of 2%, SFSHs exhibited the highest tensile strength, with a tensile stress of 192.98 ± 11.98 kPa, a tensile modulus of 18.21 ± 1.35 kPa, and a tensile strain of 1170.98 ± 68.76%. Visually, the resultant hydrogel could be stretched 10 times of its original length with or without being twisted or knotted and was stiff enough to withstand a massive load of up to 1 kg without any destruction (Fig. 4A).

We then compared the properties of SFSHs with conventional polyacrylamide hydrogels to disclose the underlying mechanism of the increment in mechanical strength. Similarly, polyacrylamide hydrogels crosslinked by MBAA with equivalent double bonds to OMV-AM were used as the control. As expected, SFSHs showed a typical J-shaped tensile stress-strain curve with a non-linear relationship, giving the maximal tensile stress and strain of 204.48 kPa and 1268.22%, respectively (Fig. 4B). In contrast to the control polyacrylamide hydrogel, the tensile stress, modulus, and strain of SFSHs were elevated separately by 4.39-fold, 1.08-fold, and 2.38-fold (Fig. 4C–E). Similar enhancements were observed in the study of the compression property of SFSHs. Crosslinking by OMV-AM increased the compressive performance of SFSHs as suggested by a steeper compressive stress-strain curve (Fig. 4F). At a compressive strain of 95%, the compressive stress and modulus of SFSHs approached 3865.27 ± 169.97 kPa and 26.16 ± 2.48

kPa, which were 1.31- and 1,16-times higher than those of the control hydrogel, respectively (Fig. 4G, H). The above results indicated that replacement of small molecular crosslinker with OMV-AM could strengthen the mechanical behavior of polyacrylamide hydrogels. We speculated that this improvement could be explained by the presence of deformable crosslinking point, namely the spherical surface of OMV linked with numerous polymer chains. Different from traditional crosslinking point, a chemical bond, vesicular crosslinking point could be deformed to adapt the stretching of network, affording enhanced mechanical toughness of the hydrogels. This is similar to cell-ECM network in the skin, in which cells are able to adapt stretching through deformation by virtue of the fluidity of cell membrane[44]. To verify our hypothesis by SEM imaging, the SFSHs strips were stretched to approximately 400% strain and maintained in the stretched state for freeze-drying. As given in Fig. 4I, J, OMVs in SFSHs were deformed into a flat shape in the same direction that the gel was stretched. It was further quantified that the long-to-short diameter ratio and the average long diameter of the crosslinked OMVs in lyophilized SFSHs separately increased from 1.41 ± 0.36 to 1.73 ± 0.35 and 285 ± 69 nm to 332 ± 86 nm after stretching (Fig. 4K, L). Furthermore, cyclic tensile test was performed to verify the energy dissipation of SFSHs. Ten cycles of tensile loading and unloading at a strain of 1000% were subjected to SFSHs, while a strain of 400% was conducted for the control polyacrylamide hydrogel due to the lower strain. Compared to

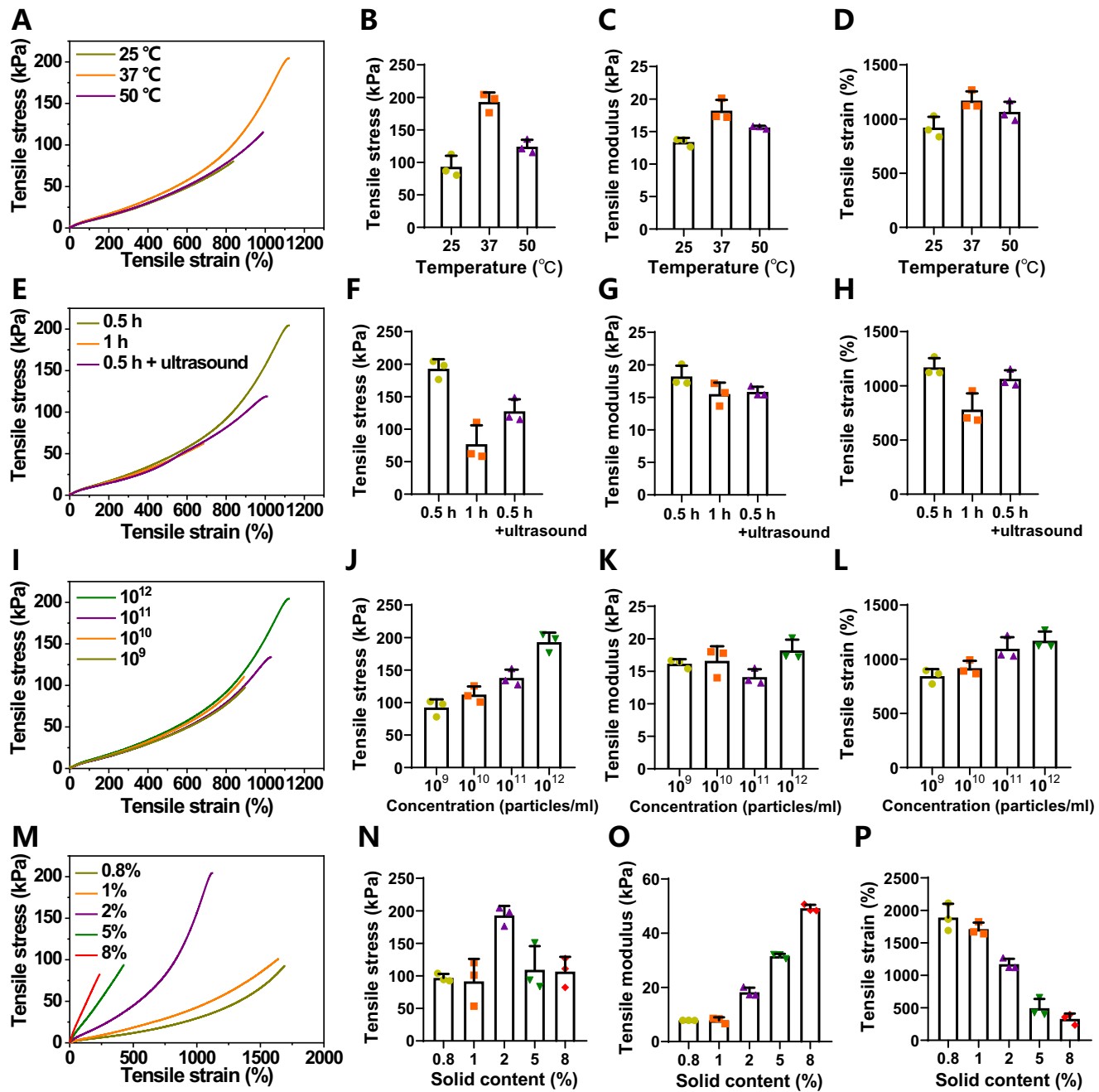

**Fig. 3 | Optimization of SFSHs. A** Tensile stress versus strain curves, **B** tensile stress, **C** tensile modulus, and **D** tensile strain of SFSHs formed by OMV-AM that were prepared at different reaction temperatures. **E** Tensile stress versus strain curves, **F** tensile stress, **G** tensile modulus, and **H** tensile strain of SFSHs formed by OMV-AM that were prepared at different reaction time and operation procedures. **I** Tensile stress versus strain curves, **J** tensile stress, **K** tensile modulus, and **L** tensile strain of SFSHs formed by OMV-AM at different particle concentrations. **M** Tensile stress versus strain curves, **N** tensile stress, **O** tensile modulus, and **P** tensile strain of SFSHs crosslinked by DSPE-PEG-AM at different solid contents. Long strip samples (25 × 4 × 2 mm) were used for the tests. Data are presented as mean values ± SD (*n* = 3, from independent experiments).

the control hydrogel, the loading and unloading curves of the first cycle for SFSHs noticeably deviated with a significant elastic hysteresis loop, and the hysteresis circle and dissipated energy of the first cycle were markedly higher than those of the tenth cycle (Supplementary Fig. 3). These results indicated that the deformation and recovery of vesicular crosslinking points within SFSHs consumed a portion of energy, leading to non-coinciding tensile loading and unloading curves. These findings validated that during loading, polymer network crosslinked by mechanoresponsive OMVs could efficiently dissipate stretching energy by vesicular deformation, leading to increments in the mechanical properties of SFSHs.

## Antibacterial ability of SFSHs

OMVs carry diverse bioactive substances derived from parent cells play critical roles in resisting or inhibiting rival bacterial species. Note that Nissle 1917 cells secrete microcin that possesses effective antibacterial ability, particularly to inhibit the proliferation of pathogenic *Salmonella Typhimurium* (STm)[45]. To assess the antibacterial ability of SFSHs, the growth of STm was monitored by co-incubation for different time intervals. Both untreated STm and STm incubated with regular polyacrylamide hydrogels were used as controls and the growth curves were obtained by recording the MFI of survived STm expressing mCherry. As expected, we found that OMVs secreted by

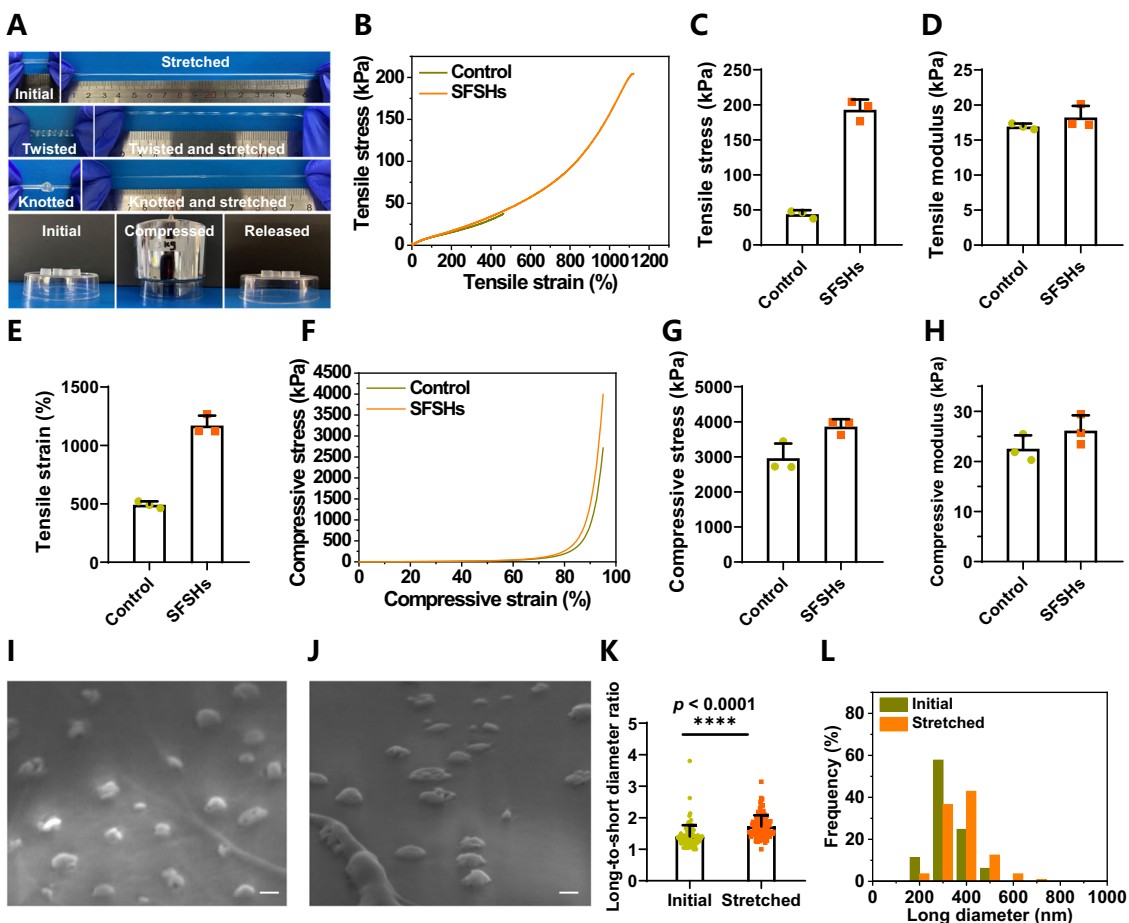

**Fig. 4 | Enhanced mechanical strength of SFSHs. A** Photographs of SFSHs under different treatments including being stretched, twisted, knotted, and compressed first and then released. **B** Tensile stress versus strain curves, **C** tensile stress, **D** tensile modulus, and **E** tensile strain of SFSHs and control polyacrylamide hydrogels. Long strip samples (25 × 4 × 2 mm) were used for the tests. **F** Compressive stress versus strain curves, **G** compressive stress, and (**H**) compressive modulus of SFSHs and control polyacrylamide hydrogels. Cylindrical samples (8 × 5 mm) were used for the tests. Data are presented as mean values ± SD ($n = 3$, from independent experiments). SEM images of lyophilized SFSHs **I** before and **J** after stretching to approximately 400% strain. Scale bar: 200 nm. **K** Long-to-short diameter ratio and **L** average long diameter of the vesicles embedded in SFSHs before and after stretching, respectively. Data are analyzed using Image J software and presented as mean values ± SD ($n \approx 100$, from independent vesicles). Statistical analysis was performed using an unpaired Student's $t$ test (two-tailed) between two groups, giving $p$ values, **** $p < 0.0001$.

Nissle 1917 cells were able to suppress the growth of STm (Fig. 5A). Similarly, SFSHs presented remarkable antibacterial ability, as evidenced by a retarded growth of STm over time. While, the control polyacrylamide hydrogel showed negligible bactericidal efficacy given that the incubated STm exhibited a near-coincident growth curve to untreated STm. The ability of SFSHs to inhibit STm growth could be simply attributed to the presence of OMVs, which inherited antibacterial microcin from Nissle 1917 cells. In addition, different from free OMVs, the growth of STm was slowed with the volume of incubated SFSHs increasing from 10% (volume/volume) to 20% (Fig. 5B and Supplementary Fig. 4), showing a dose-dependent bacteriostatic effect. Compared to free OMVs, the enhanced antibacterial capabilities of SFSHs might be attributed to the mechano-responsiveness of covalently crosslinked OMVs. During incubation, swelling-mediated network stretching could induce a less tightly arranged phospholipid structure and subsequently trigger the release of antimicrobial substances from the OMVs, thereby endowing SFSHs with enhanced antibacterial ability. The number of survived STm was further quantified by bacterial plate counting (Fig. 5C-E). Apparently, co-incubation with 22% SFSHs resulted in the lowest number of bacterial colonies, which was decreased by 55.67-times compared to untreated STm. The number of survived STm decreased to 1.79% only after 3 h co-incubation with 22% SFSHs, suggesting the potent killing ability

against pathogens. Similar to that skin-resident immune cells, e.g., T lymphocytes, can release cytotoxins to kill pathogens[46], the embedded OMVs carrying microcin could endow SFSHs with satisfactory antibacterial ability. It was anticipated that by choosing different types of OMVs, SFSHs could be introduced with diverse and even specific antibacterial activities given the versatility of DSPE-PEG-AM to functionalize OMVs.

## Immunological activity of SFSHs

As the presence of a large number of OMVs, we next studied the immunogenicity of SFSHs. As the most potent and professional antigen-presenting cells (APCs) and the only APC capable of activating naïve T cells, dendritic cells (DCs) play decisive and instructive roles in adaptive immune responses. DCs present specific antigens to T cells and provoke the massive proliferation of antigen-specific T cells which subsequently eliminate pathogens with specific antigen characteristics[47,48]. Considering that the maturation of DCs is pivotal for exerting their functions such as T cell activation and polarization, we hence investigated the immunostimulatory ability of SFSHs toward the maturation and activation of DCs. Note that DCs-induced activation of T cells is mediated by two major pathways including the binding of major histocompatibility complexes (MHC) on DCs to T-cell receptors and the conjugation between costimulatory molecules

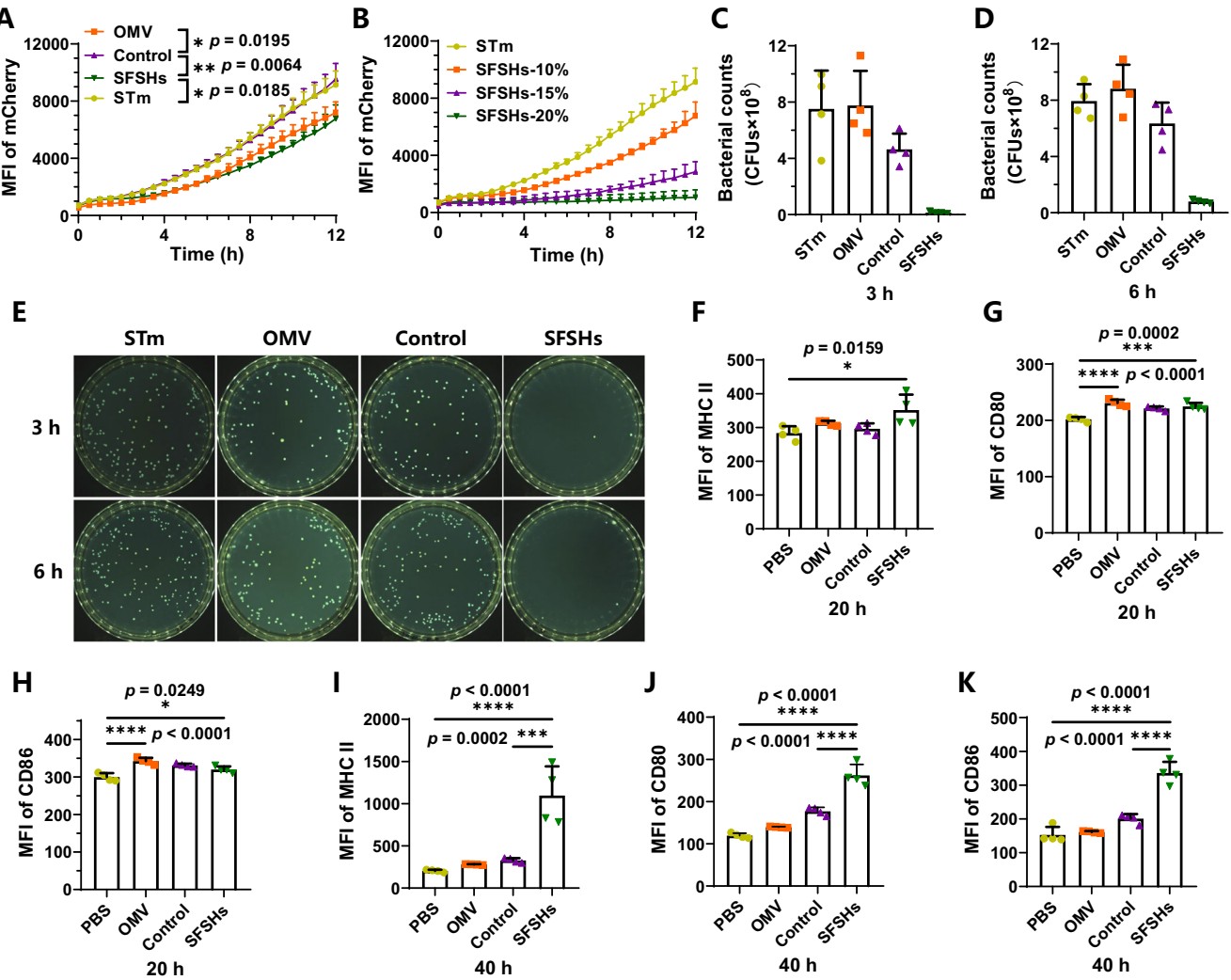

**Fig. 5 | Antibacterial ability and immunological activity of SFSHs. A** Survival curves of STm after co-incubation with free OMVs, control polyacrylamide hydrogel, and SFSHs at the same dose measured by recording the MFI of expressed mCherry. Untreated STm were used as a control. **B** Survival curves of STm after co-incubation with SFSHs at different doses. The number of survived STm after co-incubation with free OMVs, control polyacrylamide hydrogel, and SFSHs for **C** 3 h and **D** 6 h by bacterial plate counting. **E** Digital images of plates containing survived STm after co-incubation with free OMVs, control polyacrylamide hydrogel, and SFSHs for 3 h (upper) and 6 h (below). The images displayed the results of $10^6$-fold (upper) and $10^7$-fold (below) dilution. Expression levels of **F, I** MHC II, **G, J** CD80, and **H, K** CD86 on DC2.4 cells after incubation with PBS, free OMVs, control polyacrylamide hydrogel, and SFSHs for 20 h and 40 h, respectively. Data are presented as mean values ± SD ($n = 4$, from independent experiments). Statistical analysis was performed using one-way ANOVA with Tukey's multiple comparison test, giving $p$ values, * $p < 0.05$, ** $p < 0.01$, *** $p < 0.001$, **** $p < 0.0001$.

(CD80 and CD86) on DCs and CD28 on T cells[49]. The expression levels of MHC II, CD80, and CD86 were detected by FCM analysis after incubation DC2.4 cells with SFSHs for the indicated time intervals. PBS, free OMVs, and the regular polyacrylamide hydrogel were separately incubated as controls. As shown in Fig. 5F–K and Supplementary Figs. 5-8, compared to the PBS and polyacrylamide hydrogel groups, the SFSH group exhibited clear increments in the expression levels of MHC II, CD80, and CD86, while the free OMV group achieved enhancements only in CD80 and CD86 after 20 h incubation. The enhanced immunological activity of SFSHs might be ascribed to the dynamic deformation of OMVs, which could promote the release of intrinsic antigens, inflammatory mediators, and other bioactive components. The levels of MHC II, CD80, and CD86 induced by SFSHs were near 1.24-, 1.11-, and 1.07-times higher than those of PBS. With the incubation time prolonging to 40 h, the expressions of MHC II, CD80, and CD86 on DC 2.4 cells treated with SFSHs further elevated and were significantly higher than those of other groups. As calculated, the expressions of MHC II, CD80, and CD86 were respectively increased by 5.37-, 2.19-, and 2.20-fold compared to the PBS group. These results

verified that SFSHs could promote the expression of the major histocompatibility complexes MHC II and upregulate the costimulatory molecules such as CD80 and CD86 on DCs, showing the highest efficacy to activate DCs and mediate the following antigen processing and presentation among all these groups.

## Versatility of SFSHs to introduce different matrixes

Having confirmed the successful preparation of SFSHs by crosslinking OMVs, we turned our attention to explore the versatility of this approach to tune the structure and function of the resulting hydrogels. Given that cells were connected by different matrixes such as multilamellar lipids, collagen, elastin, and hyaluronic acid in the skin[50], we expanded the ability of this approach by introducing a second OMV-crosslinked network to form an interpenetrating double-network structure. As a proof-of-concept study, a second PEG network was formed by a catalyst-free azide-alkyne click reaction (Fig. 6A). Azido-modified OMVs (OMV-N$_3$) and dibenzocyclooctyne (DBCO) end-capped linear PEG (DBCO-PEG-DBCO) were chosen as the clickable crosslinker and the precursor of the second network, respectively.

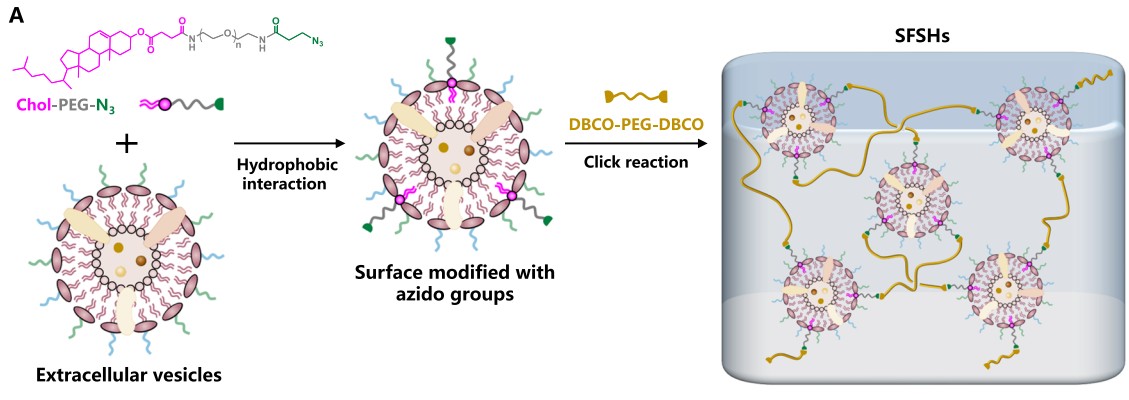

**B  Versatility to introduce different matrixes**

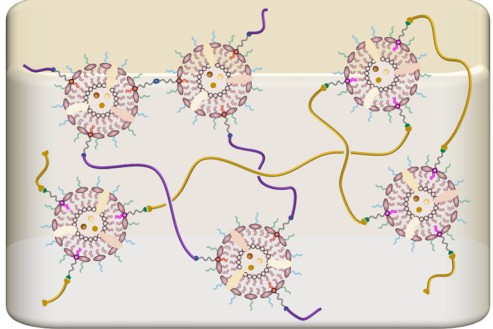

**C  Versatility to incorporate different cell-membrane compartments**

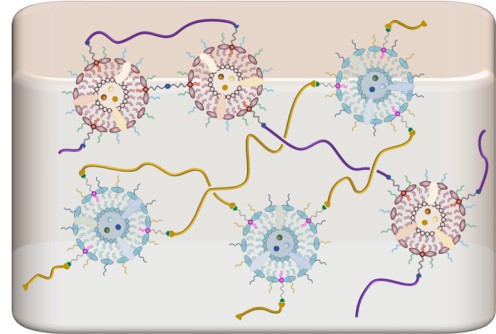

**Fig. 6 | Schematic illustration. A** Synthetic route of second PEG-OMV hydrogel. OMV-crosslinked PEG network was fabricated via catalyst-free azide-alkyne click-reaction using OMV-N$_3$ as the crosslinker and linear DBCO-PEG-DBCO as the precursor. **B** Illustration for the versatility of SFSHs to introduce different matrixes. **C** Illustration for the versatility of SFSHs to incorporate different cell-membrane compartments.

OMV-N$_3$ were fabricated via hydrophilic/hydrophobic self-assembly under a simple co-incubation of cholesterol-PEG (2 kDa)-azide (Chol-PEG-N$_3$) with OMVs at 37 °C for 1 h, while DBCO-PEG-DBCO was synthesized via amidation of DBCO acid with amine-PEG (10 kDa)-amine (NH$_2$-PEG-NH$_2$) (Supplementary Fig. 9). The successful fabrication of OMV-N$_3$ was verified by FCM and LSCM with the help of a DBCO-modified fluorescein isothiocyanate (DBCO-FITC). The fluorescence peak of OMV-N$_3$ significantly shifted to a higher intensity with 6.34 times MFI over the original OMVs (Fig. 7A, B). LSCM images visualized that OMV-N$_3$ were fluorescently labeled (Fig. 7C). OMV-N$_3$ presented a decreased average particle size of 165.8 nm and an increased zeta potential of −7.8 mV (Fig. 7D, E). Before the construction of the double-network structure, we first studied the formation of PEG-OMV network by click reaction. As expected, hydrogels were formed rapidly within minutes after mixing the aqueous solutions of DBCO-PEG-DBCO and OMV-N$_3$. Typical 3D LSCM images clearly showed that a large number of OMVs labeled with fluorescent dye FITC-PEG-NHS were uniformly dispersed within the hydrogel (Fig. 7F). The compressive stress was significantly enhanced with the increase of OMV-N$_3$ particle concentration (Fig. 7G-I). Note that the hydrogel crosslinked by ~1.5 × 10$^{12}$ particles/ml OMV-N$_3$ displayed the highest compressive stress of 473.11 ± 98.78 kPa and the highest compressive modulus of 260 ± 89.81 Pa at the compressive strain of 90%. We then constructed the polyacrylamide-OMV/PEG-OMV double-network hydrogel (DN$_1$) by a simple one-pot synthesis procedure in light of the orthogonal feature between radical polymerization and click reaction (Fig. 6B). As visualized by 3D LSCM imaging, both Cy5-NHS labeled OMV-AM (red) and FITC-PEG-NHS labeled OMV-N$_3$ (green) were present in the hydrogel (Fig. 7J), indicating the formation of both OMV-AM/polyacrylamide and OMV-N$_3$/DBCO-PEG-DBCO networks.

The resultant DN$_1$ emerged a greatly enhanced compressive stress-strain behavior compared to the OMV-AM/polyacrylamide

single-network hydrogel (SN, Fig. 7K–M). The compressive stress and modulus were 2813.72 ± 316.69 kPa and 17.71 ± 5.61 kPa, which were increased by 1.52- and 6.73-times, respectively. In addition, DN$_1$ exhibited improved lubrication property compared to SN, as evidenced by the prolonged sliding distances on the surfaces of different substrates including glass, polystyrene (PS), aluminum (Al), and poly-tetrafluoroethylene (PTFE) (Fig. 7N). Particularly, DN$_1$ achieved a remarkable sliding distance of up to 110 mm on the PTFE surface, while no slippage occurred to SN. This anti-adhesive characteristic was further verified by significantly reduced adhesion of proteins on the surface (Fig. 7O). After incubation with 150 μg/ml FITC-conjugated bovine serum albumin (BSA-FITC) protein solution for 10 min, the intensity of fluorescence on DN$_1$ was decreased by approximately 4.75-times compared to that of SN (Fig. 7P). Similarly, after incubation with NIH/3T3 cells for 24 h, the number of adhered cells on DN$_1$ was substantially reduced (Fig. 7Q). These anti-adhesive and antifouling properties were attributed to the hydrophilicity and low surface energy of PEG network[51].

## Versatility of SFSHs to incorporate different cell-membrane compartments

On the other hand, given the fact that different types of cell-membrane compartments were embedded in the skin, we also expanded the versatility of this method by developing a double-network structure that was crosslinked by two different types of OMVs (Fig. 6C). OMVs derived from *Staphylococcus epidermidis* (SE) cells were collected and modified with azido groups (OMV$_{SE}$-N$_3$). As shown in Supplementary Fig. 10, both FCM and LSCM results confirmed the successful fabrication of OMV$_{SE}$-N$_3$, which possessed an enlarged average particle size of 195.4 nm and an increased zeta potential of −4.0 mV. Meanwhile, OMVs secreted by Nissle 1917 cells were functionalized with double bonds (OMV$_{EcN}$-AM). The hydrogel consisting of both polyacrylamide-

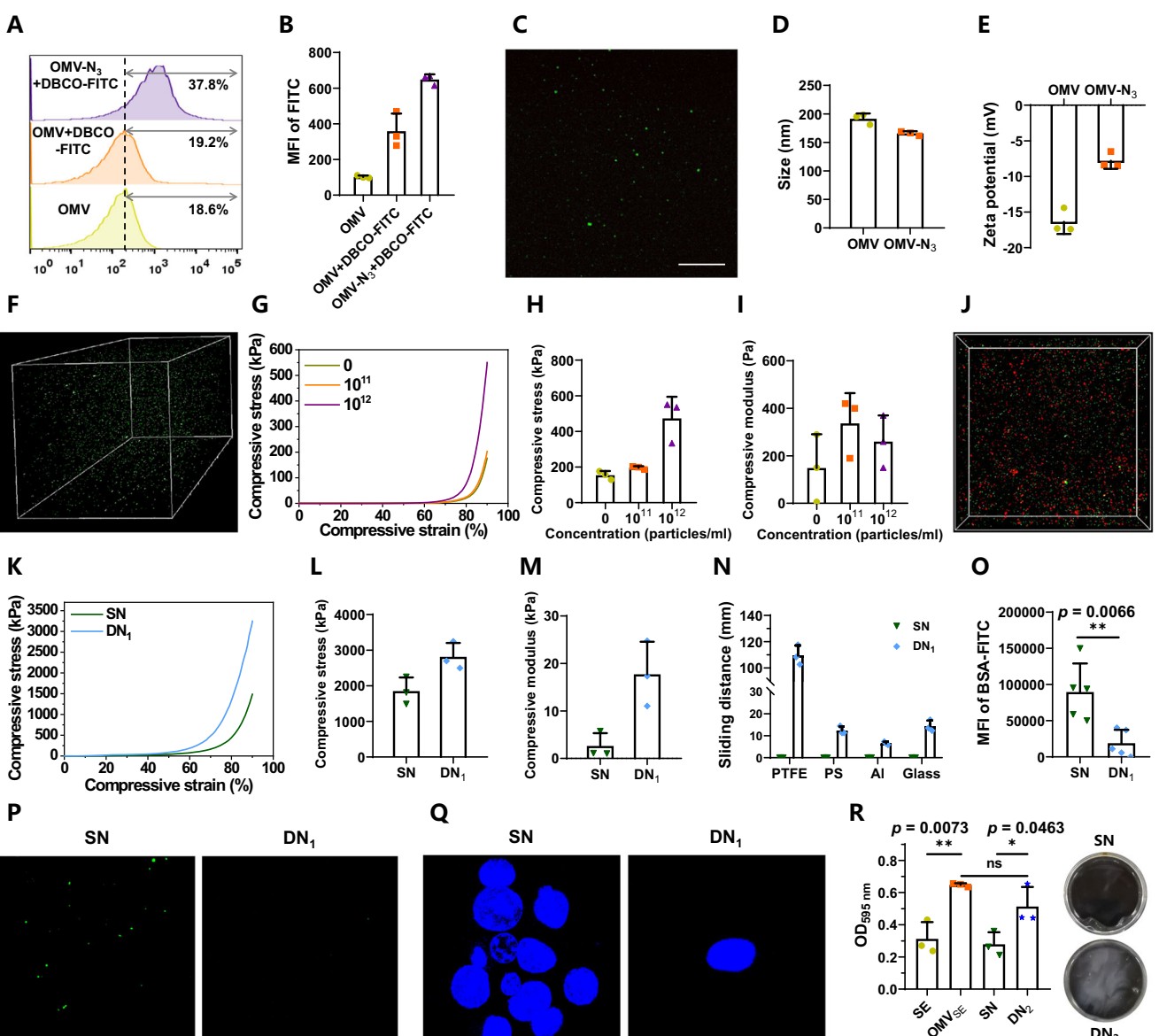

**Fig. 7 | Versatility of SFSHs. A** FCM histograms of OMV and OMV-N$_3$ after co-incubation with DBCO-FITC at 25 °C for 0.5 h. **B** MFI values of FITC-labeled OMV and OMV-N$_3$. **C** LSCM imaging of FITC-labeled OMV-N$_3$. Scale bar: 25 μm. **D** Average size and **E** zeta potential of OMVs and OMV-N$_3$. **F** 3D LSCM imaging of PEG-OMV hydrogel with FITC-labeled OMVs. **G** Compressive stress versus strain curves, **H** compressive stress, and **I** compressive modulus of PEG-OMV hydrogel containing different particle concentrations of OMV-N$_3$. **J** 3D LSCM imaging of DN$_1$ derived from Cy5-labeled OMV-AM and FITC-labeled OMV-N$_3$. **K** Compressive stress versus strain curves, **L** compressive stress, and **M** compressive modulus of SN and DN$_1$. Cylindrical samples (8 × 5 mm) were used for the compression tests. **N** Sliding distances of SN and DN$_1$ on different substrates. **O** MFI values and **P** LSCM images of SN and DN$_1$ after incubation with BSA-FITC (150 μg/ml) for 10 min. Scale bar: 10 μm. **Q** LSCM images of SN and DN$_1$ after incubation with NIH/3T3 cells for 24 h. Blue: nuclei (Hoechst 33342). Scale bar: 10 μm. **R** Biofilm formation of SE after co-incubation with PBS, free OMV$_{SE}$, SN, or DN$_2$ for 24 h. Top and bottom photographs separately indicate biofilm formation treated by SN and DN$_2$. Data are presented as mean values ± SD (*n* = 3 or 5, from independent experiments). Statistical analysis was performed using an unpaired Student's *t* test (two-tailed) between two groups or a one-way ANOVA with Tukey's multiple comparison test among multiple groups, giving *p* values, * *p* < 0.05, ** *p* < 0.01. ns, no significance.

OMV$_{EcN}$ and PEG-OMV$_{SE}$ networks (DN$_2$) were fabricated under the same condition as previously described. Note that SE is a commonly existing commensal bacterium on human skin[52]. We hence speculated that double network hydrogels formed by SE-derived vesicles could facilitate the colonization of symbiotic bacteria on the surface. Given that biofilm formation is a critical mechanism for bacterial colonization[53], we evaluated the biofilm-forming ability of SE on DN$_2$. After incubation SE with PBS, free OMV$_{SE}$, polyacrylamide-OMV$_{EcN}$ hydrogel (SN), and DN$_2$ for 24 h, the optical density (OD) quantity of formed biofilm was measured at 595 nm by using a microplate reader. As expected, OMV$_{SE}$ significantly enhanced biofilm formation compared to untreated SE, showing a 2.07-fold increment (Fig. 7R).

Interestingly, DN$_2$ exhibited comparable biofilm-promoting capability to OMV$_{SE}$, being 1.85-times higher than SN. This increment was supported by direct observation of the culture dishes, displaying extensive biofilm formation at the bottom of the dishes in the DN$_2$-treated group instead of the SN group. The enhancement in biofilm formation could be explained by the presence of OMV$_{SE}$, which inherited massive bioactive substances from the parent SE cells[54]. These findings validated the versatility of the cell-membrane compartment crosslinking approach to adjust the structure and functionality of SFSHs.

In summary, we have proposed a cell-membrane compartment crosslinking strategy to develop bioinspired hydrogels with both structurally and functionally skin-like characteristics. To prepare the

crosslinked cell-membrane-based network, we employ a free radical aqueous polymerization of acrylamide using extracellular vesicles decorated with olefinic double bonds as a crosslinker. In comparison to conventional methylene diacrylamide-crosslinked polyacrylamide hydrogels, SFSHs exhibit significantly enhanced mechanical strength. This is attributed to the attachment of multiple polymer chains to a mechano-deformable compartment crosslinker, namely the double bond-functionalized vesicle, which allows for energy dissipation through vesicular deformation. SFSHs also show specific antibacterial effects and demonstrate a potent capability to induce the maturation and activation of dendritic cells due to the existence of massive bioactive substances in the extracellular vesicles. To showcase the versatility of this methodology, we further verify its ability to control the structure and function of SFSHs by incorporating a second network. This is achieved through catalyst-free click reaction-enabled crosslinking between alkyne-double-ended PEG and azido-modified extracellular vesicles. Given the flexibility of this approach, we anticipate that this biomimetic paradigm paves an avenue for the preparation of advanced skin-inspired biomaterials.

## Methods

### Materials and strains
*Escherichia coli* Nissle 1917 (EcN), *Salmonella typhimurium* (STm), and *Staphylococcus epidermidis* (SE) were procured from China General Microbiological Culture Collection Center (Beijing, China). DC2.4 cells (a murine dendritic cell line) were purchased from Millipore Sigma (SCC142) and NIH/3T3 cells (a murine fibroblast cell line) were obtained from American Type Culture Collections (ATCC CRL-1658). Fluorescent dyes $N$-hydroxysuccinimide (NHS) ester-functionalized cyanine5 (Cy5) (Cy5-NHS), NHS ester-functionalized fluorescein isothiocyanate (FITC) (FITC-PEG-NHS), dibenzocyclooctyne (DBCO)-modified FITC (DBCO-FITC), sulfhydrated Cy5 (SH-Cy5), and Hoechst 33342 were purchased from Lumiprobe Corporation. Distearoylphosphatidylethanolamine-polyethylene glycol (2 kDa)-acrylamide (DSPE-PEG-AM) and cholesterol-polyethylene glycol (2 kDa)-azide (Chol-PEG-N$_3$) were provided by Shanghai Ponsure Biotech, Inc. Acrylamide (99%), $N$, $N'$-methylenebisacrylamide (MBAA), ammonium persulfate (APS, 99.99%), $N$, $N$, $N'$, $N'$-tetramethylethylenediamine (TEMED, 99%), 2, 2'-azobis(2-methylpropionitrile) (AIBN, ≥ 95%), O-(7-azabenzotriazol-1-yl)-$N$, $N$, $N'$, $N'$-tetramethyluronium hexafluorophosphate (HATU, 99%), and $N$, $N$-diisopropylethylamine (DIEA, ≥ 99%) were supplied by Sigma. Dibenzocyclooctyne-acid (DBCO acid, ≥ 95%) was obtained from Xi'an ruixi Biological Technology Co., Ltd. Amine-polyethylene glycol (10 kDa)-amine (NH$_2$-PEG-NH$_2$) was purchased from Xiamen Sinopeg company. Plasmid pBBR1MCS2-Tac-mCherry (Kanamycin resistant) and phosphate buffer saline (PBS, 1 ×, pH 7.4) were provided by Sangon Biotech company. Anhydrous $N$, $N$-dimethylformamide (DMF, 99.8%) and ethyl ether (Et$_2$O, ≥ 99.7%) were supplied by Sinopharm chemical reagent Co., Ltd. Other reagents and solvents were used as received.

### Purification and characterization of OMVs
EcN were cultured in LB medium at 37 °C with 100 g/ml kanamycin. After overnight incubation, the medium was diluted 100-fold using fresh LB medium with 100 g/ml kanamycin and 3 g/ml ampicillin, and continued culturing for additional 2 days. After centrifugation at a speed of 7600 g for 30 min, the resulting supernatant was filtered through 0.45-μm filters at 4 °C to remove bacterial cells. The filtrate was then pelleted by ultracentrifugation (170,000 g, 4 °C for 1 h). OMVs were washed once with sterile PBS and concentrated to 1 ml by ultracentrifugation (170,000 g, 4 °C for 1 h). Finally, the concentrated solution was stored at −80 °C for further experiments. The average size and zeta potential of OMVs were measured by DLS (Malvern Zetasizer Nano ZS, UK). The number concentration of OMVs was determined by NTA test (Malvern NanoSight NS300, UK). The total protein

concentration of concentrated OMVs was determined using BCA assay (Beyotime, China). The morphology of OMVs was characterized by TEM (Hitachi, Japan). OMV samples were negatively stained with 1% (w/v) phosphotungstic acid for 1 min before TEM imaging. For visualization, OMVs were labeled with Cy5-NHS (red) or FITC-PEG-NHS (green) as needed and presented by LCSM (Leica TCS SP8, Germany) with a 63× oil objective.

### Preparation and characterization of OMV-AM
For the surface modification of OMVs with olefinic double bonds, concentrated OMVs were incubated with DSPE-PEG-AM with a solid content of 2% at 37 °C under shaking (500 g) for 0.5 h. FCM (CytoFLEX, Beckman Coulter, USA) and LSCM analyses were performed to verify whether the modification was successful. The resultant OMV-AM were coincubated with AIBN and SH-Cy5 at 37 °C under shaking (500 g) for 3 h. The size and zeta potential variations of OMV-AM were measured by DLS. The morphology of OMV-AM was determined by TEM.

### Preparation and optimization of SFSHs
Using OMV-AM solution as a solvent, acrylamide (284 mg, 4 mmol), initiator APS (5‰ w/w), and catalyst TEMED (5‰ w/w) were added successively to give a pre-gel solution at room temperature in a nitrogen glove box. After 30 min polymerization in PTFE molds, SFSHs were obtained and ready for further measurements. Polyacrylamide hydrogel that was crosslinked by the small molecular crosslinker MBAA without the addition of OMVs was used as control. For optimization of preparation conditions, the reaction conditions of OMV-AM including temperature (25 °C, 37 °C, or 50 °C), time (0.5 h or 1 h), and operation procedures (with or without ultrasound) for the modification of OMVs, and the crosslinking density of the polymer network encompassing particle number of vesicles ($1 \times 10^9$ to $1 \times 10^{12}$) and the density of double bonds decorated on OMVs (solid content of DSPE-PEG-AM from 0.8% to 8%), were systematically studied. For visualization, SFSHs were fabricated by OMV-AM labeled with Cy5-NHS and observed by 3D LCSM imaging.

### SEM analysis of SFSHs
The microstructure and morphology of lyophilized SFSHs with different OMV-AM concentrations were observed by SEM (Zeiss 1550VP FESEM, Germany) under an accelerating voltage of 3 kV. For the verification of vesicular crosslinking point, the samples were stretched to 3 times their initial length before lyophilization. All the samples were gold-sputtered prior to the tests.

### Mechanical characterization
All the mechanical tests of hydrogels were conducted on the universal material testing machine (Instron-3342, US). Compression tests were performed at a strain rate of 5 mm/min and the dimension of the cylindrical samples was 8 × 5 mm, and the compression strain was set at 90% or 95%. Tensile tests were carried out under a strain rate of 100 mm/min and the dimension of the rectangle samples was 25 × 4 × 3 mm. Three parallel duplicates were employed for each test.

### Antibacterial ability of SFSHs
STm expressing mCherry were cultured overnight in LB medium with 100 μg/ml kanamycin at 37 °C under shaking (170 g). Bacteria were collected and washed with PBS and then diluted to reach an OD value at 600 nm of -1 in LB. Afterward, 50 μl bacterial suspension and SFSHs were added into a 96-well plate, additional LB medium with 100 μg/ml kanamycin was finally added until reaching 200 μl. The mixtures were co-incubated at 37 °C with gently shaking for 12 h. The MFI at 590 nm of live STm expressing mCherry was recorded at 0.5-h intervals with a microplate reader (HIMF, BioTek, USA). Untreated STm and STm treated with free OMVs or regular polyacrylamide hydrogels at the same dose were used as controls. The effects of free OMVs or SFSHs at

different doses (10%, 15%, or 20%, volume/volume) were also studied. The number of survived STm was further quantified by bacterial plate counting. 50 μl STm ($OD_{600}$ = 1) and 300 μl SFSHs were added in 1 ml LB medium and cultured at 37 °C under shaking (170 g). A series of 50-μl droplets were withdrawn at 3 h and 6 h, which were spread on LB agar plates with 100 g/ml kanamycin and further incubated at 37 °C overnight for bacterial counting. Each sample was performed in three parallel.

## Immunological activity of SFSHs

The effects of SFSHs on the maturation and activation of dendritic cells were investigated in vitro. DC2.4 cells were cultured in RPMI 1640 medium supplemented with 10% (v/v) fetal bovine serum (FBS) and 1% (v/v) antibiotics (penicillin-streptomycin, PS) in a cell incubator (37 °C, 5% $CO_2$). Cells were seeded in 24-well plates at a density of $10^5$ cells/well, following the treatment of PBS (10 μl), free OMVs (10 μl), regular polyacrylamide hydrogels (10 μl), or SFSHs (10 μl) for 20 h and 40 h, respectively. Then, cells were collected and washed with PBS. For specific labeling, cells were stained with anti-mouse antibody against CD86-APC (GL-1, Cat: 105012, BioLegend), CD80-PE (16-10A1, Cat: 104708, BioLegend), I-A/I-E (MHC II)-PE/Cy7 (M5/114.15.2, Cat: 107629, BioLegend) in 0.5% bovine serum albumin (BSA) dissolved in PBS on ice for 1 h. Antibodies were used at a dilution of 1:200. After washing with PBS for 3 times, cells were analyzed by FCM.

## Preparation and characterization of OMV-$N_3$

For the surface functionalization of OMVs with azido groups, concentrated OMVs were incubated with Chol-PEG-$N_3$ (19 μM) at 37 °C under shaking (500 g) for 1 h. FCM and LSCM analyses were performed to confirm the successful functionalization. The obtained OMV-$N_3$ were incubated with DBCO-FITC at 25 °C under shaking (500 g) for 0.5 h. The size and zeta potential of OMV-$N_3$ were measured by DLS.

## Preparation and characterization of PEG-OMV hydrogels

Using OMV-$N_3$ solution as a solvent, DBCO-PEG-DBCO was added in the stoichiometric proportion (molar ratio, 1:1) according to the amount of Chol-PEG-$N_3$. PEG-OMV hydrogels were obtained in PTFE molds within minutes at room temperature. For visualization, PEG-OMV hydrogels were fabricated by OMV-$N_3$ labeled with FITC-PEG-NHS and observed by 3D LCSM. Compression tests were performed according to the method described previously.

## Construction and characterization of $DN_1$

Using OMV-AM and OMV-$N_3$ mixed solution as a solvent, acrylamide (284 mg, 4 mmol), APS (5‰ w/w), DBCO-PEG-DBCO, and TEMED (5‰ w/w) were added successively to give a pre-gel solution at room temperature in a nitrogen glove box. After 30 min polymerization in PTFE molds, polyacrylamide-OMV/PEG-OMV double network hydrogels ($DN_1$) were obtained. DBCO-PEG-DBCO was added in the stoichiometric proportion (molar ratio, 1:1) according to the Chol-PEG-$N_3$ amount. For visualization, OMV-AM and OMV-$N_3$ were labeled respectively with Cy5.5-NHS and FITC-PEG-NHS prior to the fabrication of hydrogels and observed by 3D LCSM. Compression tests were also conducted.

## Measurement of antifouling abilities

Lubricating performance and anti-adhesion property against protein and cells were studied to evaluate the antifouling abilities of hydrogels. To determine lubricating performance, cylindrical samples (5 × 5 mm) were positioned on a vertical substrate. The sliding distances of polyacrylamide-OMV single network hydrogels (SN) and $DN_1$ on different substrates including glass, PS, Al, and PTFE were recorded and repeated three times. To analyze anti-protein adhesion, SN and $DN_1$ were incubated with BSA-FITC solution (150 μg/ml) for 10 min. After washing twice with PBS, samples were observed by LCSM, and the MFI

of adhesive BSA-FITC was recorded at 520 nm with a microplate reader. To measure anti-cell adhesion, NIH/3T3 fibroblast cells were cultured in Dulbecco's modified Eagle's medium (DMEM) with 10% FBS and 1% PS in a cell incubator (37 °C, 5% $CO_2$). SN and $DN_1$ were placed in the confocal dish. NIH/3T3 cells were seeded at a density of $5 × 10^5$ cells and incubated for 24 h. Medium and unattached cells were removed. Samples were stained with 10 μg/ml Hoechst 33342 for 10 min and washed three times and then imaged by LCSM.

## Enhanced biofilm formation of SE

SE was grown at 37 °C overnight in LB medium with 5 μg/ml erythromycin. The culture was then diluted 200 folds using fresh LB medium for further use. OMVs were collected from SE and modified with azido groups ($OMV_{SE}$-$N_3$). OMVs secreted by Nissle 1917 cells were functionalized with double bonds ($OMV_{EcN}$-AM). Both $OMV_{SE}$-$N_3$ and $OMV_{ECN}$-AM were employed to prepare polyacrylamide-$OMV_{ECN}$/PEG-$OMV_{SE}$ double network hydrogels ($DN_2$). 200 μl of SE suspension was transferred into 96-well plates, and 10 μl of free $OMV_{SE}$, polyacrylamide-$OMV_{ECN}$ hydrogel (SN), and $DN_2$ were added respectively. The mixtures were statically incubated for 24 h at 37 °C. Afterward, the supernatants were removed gently and the plates were washed twice with PBS and dried at 37 °C. 200 μl of methanol was added and incubated for 10 min. The methanol was removed and the plates were dried again. 100 μl of 1% crystal violet was transferred into each well and stained for 30 min and then washed ten times with PBS. Finally, 200 μl of 33% glacial acetic acid was added to each well and the OD at 595 nm of each well was measured by a microplate reader.

## Statistical analysis

All statistical analysis was performed using GraphPad Prism 8.0. The results were expressed as mean values ± standard deviation (SD). Statistical analysis was assessed by an unpaired Student's $t$ test between two groups or a one-way analysis of variance (ANOVA) with Tukey's multiple comparison test among multiple groups, giving $p$ values. Significance was defined as a value of $p < 0.05$, * $p < 0.05$, ** $p < 0.01$, *** $p < 0.001$, **** $p < 0.0001$. ns, no significance.

## Reporting summary

Further information on research design is available in the Nature Portfolio Reporting Summary linked to this article.

# Data availability

All data needed to evaluate the conclusions in the paper are present in the paper and/or the Supplementary Information. Data is available from the authors on request.

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

## Acknowledgements

This work was financially supported by the National Key Research and Development Program of China (2021YFA0909400, J.L.), the National Natural Science Foundation of China (22105123, F.W.; 22375127, Y.P.), the Shanghai Rising-Star Program (23QA1408600, F.W.), the Explorer Program of the Science and Technology Commission of Shanghai Municipality (21TS1400400, J.L.), the Interdisciplinary Program of Shanghai Jiao Tong University (YG2021QN35, F.W.), the Innovative Research Team of High-Level Local Universities in Shanghai (SHSMU-ZDCX20210900, J.L.; SHSMU-ZDCX20210700, Y.P.), the Science and Technology Commission of Shanghai (20DZ2270800, Y.P.), the Foundation of National Infrastructures for Translational Medicine (Shanghai) (TMSK-2021-123, Y.P.; TMSK-2021-119, J.L.), and the Two-hundred Talent (20191820, Y.P.; 20181704, J.L.).

## Author contributions

J.L. and Y.P. supervised the project. J.L., Y.P., F.W., and J.W. conceived and designed the experiments. F.W., Y.R., W.L., X.L., X.W., C.W., and Z.C. performed all experiments. All authors analyzed and discussed the data. F.W., Y.P., and J.L. wrote the paper.

## Competing interests

The authors declare no competing interests.
