## [Peer Review File · Nature Communications]

Generating dual structurally and functionally skin-mimicking hydrogels by crosslinking cell-membrane compartmentsREVIEWER COMMENTS

Reviewer #1 (Remarks to the Author):

In this manuscript, Wu et al. reported here a unique cell-membrane compartment-crosslinking strategy for constructing biomimetic hydrogels. This approach not only imitates the structural features of skin but also exhibits enhanced mechanical strength, antimicrobial properties, and immunogenicity akin to skin. Furthermore, the authors have explored the versatility of this strategy for enhancing the structure and functionality of hydrogels by introducing a second cell-membrane compartment-crosslinked network. Overall, the work is interesting and the data are solid to support the conclusions. Prior to publication, there are several minor issues that need to be addressed appropriately.

1. The resolution of confocal imaging in the manuscript is a bit low. Replacing with more distinct confocal images, such as Figure 7J, could significantly enhance the quality of the article.

2. The authors have introduced a second type of OMVs, specifically OMVSE from *Staphylococcus epidermidis* (SE) cells, and subjected them to surface modification to validate the versatility of the method. However, there is a lack of relevant characterization in the manuscript. This should be added.

3. In terms of mechanical testing, it is necessary to provide more details, including sample dimensions, testing conditions, the speed of compression or tension, and the extent of strain.

4. The authors have employed two different types of vesicles to construct skin-mimicking hydrogels, demonstrating antimicrobial and biofilm-promoting capabilities, respectively. This seems contradictory and should be clarified.

5. The authors utilize supramolecular interactions, assisted by DSPE, to carry out surface modification on OMVs. What is the underlying mechanism for this approach? This should be explained.

6. "The enhancement in biofilm formation could be explained by the presence of OMVSE, which inherited massive bioactive substances from the parent SE cells." Proper citation of relevant references is needed to support this claim. Besides, the references do not adhere to the required journal formatting.

7. There are several grammatical mistakes in the writing. A comprehensive grammar check throughout the manuscript is required. Language needs to be further polished.

"The multiple interaction between ..." should be "The multiple interactions between..."

"synthetic strategies capable to construct ..." should be "synthetic strategies are capable to construct..." or "synthetic strategies capable of constructing..."

"SFSHs was prepared via free radical ..." should be "SFSHs were prepared via free radical..."

Reviewer #2 (Remarks to the Author):

The authors have developed polyacrylamide hydrogels incorporated with cell-derived outer membrane vesicles (OMV) using click chemistry. The sections detailing the materials are well composed, effectively demonstrating the integration of OMVs and their role in modulating mechanical properties. The design of this hydrogel system is novel. The versatility in tuning the hydrogel's functionality using various cell-derived outer membrane vesicles and click chemistry combinations augments its potential applications. Subsequent evaluations of the hydrogels' antibacterial and immunostimulatory functions have been presented. However, a notable observation is the predominant reliance on flow cytometry data in discussing most of the biological results. Instead of using Mean Fluorescence Intensity (MFI) values, % of expressions should be shown with representative histograms or dot plots. Furthermore, the observed significant enhancement in immunological activity when OMVs were incorporated into the hydrogel, as opposed to the activity of standalone OMVs, is an intriguing result. However, the discussions supporting the authors' biological claims appear overly simplified. A more exploration of these findings could enrich the manuscript and provide readers with deeper insights into the implications of these results.

- Figure 4 I and J: The figures depict the hydrogel before and after full stretching. However, the strain value isn't specified in the result section. If the hydrogel was stretched, one would expect a reduced number of OMVs. As presented, the figure might not accurately represent the change.
- Figure 5 C, D, E & Figure 5 I, J, K: A detailed discussion is needed to elucidate the factors that enhance the functionality of incorporated OMVs in the hydrogel system.
- Flow Cytometry Data: Much of the biological data derives from flow cytometry. Histograms or dot plots should be included.
- Conclusion: If the intention is to emphasize the energy dissipation function of the hydrogel platform, it would be advantageous to perform cyclic tensile or compressive tests on hydrogels. Demonstrating that the loading curve does not overlap with the unloading curve would further validate energy dissipation claims.

Reviewer #3 (Remarks to the Author):

Reviewer #4 (Remarks to the Author):

Wu and colleagues generated dual structurally and functionally skin-mimicking hydrogels by cross-linking cell-membrane compartments and demonstrated their elasticity, antibacterial, and immunological activity in vitro. This study is interesting but requires significant revisions.

Specific Comments

1. "Why were only *Escherichia coli* Nissle 1917 cells used to generate outer membrane vesicles (OMVs)? This requires appropriate justification."
2. Cells are removed by low-speed centrifugation, and in some cases several differential centrifugation steps are also performed to remove cell debris, large protein and membrane aggregates (<https://doi.org/10.1002/pmic.200700196> and <https://doi.org/10.1002/mas.20175>). Such steps were not followed by this study; the purity of OMVs is questionable, as they may not be free of cellular debris, large proteins, and membrane aggregates.
3. In this study, the author used the ultracentrifugation method, which may contribute to non-OMV-associated proteins, unlike ultrafiltration techniques. In ultrafiltration, a bacterial supernatant is passed through a membrane with a given molecular weight cut-off, usually 50–100 kDa, which further questions the purity of OMVs.
4. No purification of OMVs was performed after ultracentrifugation, such as iodixanol (OptiPrep™) density gradient centrifugation or sucrose density gradient.
5. I would suggest that the authors include TEM images with multiple OMVs in figure 2D, or a single OMV that is appropriate to the average size of OMVs.
6. Why didn't the author show TEM images of OMVs and OMV-AMs?.
7. I would suggest that the authors clarify the changes in the zeta potential of OMVs and OMV-AMs.
8. Why was TEMED used to generate skin-mimicking hydrogels, even though it may be harmful upon contact with the skin?
9. Why was only *Salmonella Typhimurium* used to assess the antibacterial ability of SFSHs
10. Importantly, this study lacks in vivo testing of their skin-mimicking hydrogels

Reviewer #1

In this manuscript, Wu et al. reported here a unique cell-membrane compartment-crosslinking strategy for constructing biomimetic hydrogels. This approach not only imitates the structural features of skin but also exhibits enhanced mechanical strength, antimicrobial properties, and immunogenicity akin to skin. Furthermore, the authors have explored the versatility of this strategy for enhancing the structure and functionality of hydrogels by introducing a second cell-membrane compartment-crosslinked network. Overall, the work is interesting and the data are solid to support the conclusions. Prior to publication, there are several minor issues that need to be addressed appropriately.

Response: We thank the reviewer very much for her/his positive feedback. The manuscript has been revised accordingly.

1. The resolution of confocal imaging in the manuscript is a bit low. Replacing with more distinct confocal images, such as Figure 7J, could significantly enhance the quality of the article.

Response: As per the recommendation of the reviewer, we have replaced it with a high-resolution confocal image in the revised manuscript, as shown in Figure 7J on page 31.

2. The authors have introduced a second type of OMVs, specifically OMV_{SE} from Staphylococcus epidermidis (SE) cells, and subjected them to surface modification to validate the versatility of the method. However, there is a lack of relevant characterization in the manuscript. This should be added.

Response: We thank the reviewer for drawing our attention to this point. We have supplemented the relevant characterization of OMV_{SE} and azido group-modified OMV_{SE} (OMV_{SE}-N₃). As shown in Figure S9, both FCM and LSCM results confirmed the successful fabrication of OMV_{SE}-N₃, which possessed an increased average particle size of 195.4 nm and an enhanced zeta potential of -4.0 mV. The

relevant details have been added on page 17 in the revised manuscript and on page 15 in the revised supporting information.

3. In terms of mechanical testing, it is necessary to provide more details, including sample dimensions, testing conditions, the speed of compression or tension, and the extent of strain.

Response: We agree with the reviewer's comments. In the Methods section, we have elucidated the details of mechanical testing, which have been highlighted on page 4 in the revised supporting information.

4. The authors have employed two different types of vesicles to construct skin-mimicking hydrogels, demonstrating antimicrobial and biofilm-promoting capabilities, respectively. This seems contradictory and should be clarified.

Response: We thank the reviewer for highlighting this. We initially employed OMV_{ECN} derived from Nissle 1917 cells to construct SFSHs. Due to the inhibitory effect of Nissle 1917 cells on pathogenic *Salmonella Typhimurium* by secreting microcin, as well as the inheritance of various bioactive substances from the parental cells by OMV_{ECN}, the resulting SFSHs exhibited specific capability to suppress pathogens. Subsequently, we expanded the structure and functionality of SFSHs by incorporating different cell-membrane compartments. The introduction of OMV_{SE}, which carried massive bioactive substances derived from parental skin commensal bacterium, *Staphylococcus epidermidis*, endowed SFSHs with the ability to facilitate commensal bacterial colonization and promote biofilm formation. These seemingly contradictory performances were attributed to the introduction of distinct cell-membrane compartments that inherited different bioactive substances from the parent cells, which in turn validated the versatility of our strategy in modulating the structure and functionality of SFSHs by incorporating different cell-membrane compartments. The relevant details have been highlighted in the revised version (pages 11, 12 and 17, 18).

5. The authors utilize supramolecular interactions, assisted by DSPE, to carry out surface modification on OMVs. What is the underlying mechanism for this approach? This should be explained.

Response: The lipid bilayer structure of OMVs can undergo perturbation and loosen the compact phospholipid arrangement under specific conditions, such as appropriate temperatures or mechanical disturbance. At this point, lipid molecules with amphiphilic characteristics, such as DSPE, can leverage supramolecular hydrophobic/hydrophilic interaction to insert their hydrophobic tail into the bilayer of OMVs, thus presenting functionalized hydrophilic head on the surface of OMVs, thereby achieving surface modification of OMVs. The relevant explanation has been added on page 6 in the revised manuscript.

6. “The enhancement in biofilm formation could be explained by the presence of OMV_{SE}, which inherited massive bioactive substances from the parent SE cells.” Proper citation of relevant references is needed to support this claim. Besides, the references do not adhere to the required journal formatting.

Response: According to the reviewer’s suggestions, relevant references have been cited, and the format of the references has been adjusted to comply with the journal’s requirement.

7. There are several grammatical mistakes in the writing. A comprehensive grammar check throughout the manuscript is required. Language needs to be further polished.

“The multiple interaction between ...” should be “The multiple interactions between...”

“synthetic strategies capable to construct ...” should be “synthetic strategies are capable to construct...” or “synthetic strategies capable of constructing...”

“SFSHs was prepared via free radical ...” should be “SFSHs were prepared via free radical...”

Response: Thank you. A comprehensive grammar check and language refinement have been performed throughout the revised manuscript.

Reviewer #2

The authors have developed polyacrylamide hydrogels incorporated with cell-derived outer membrane vesicles (OMV) using click chemistry. The sections detailing the materials are well composed, effectively demonstrating the integration of OMVs and their role in modulating mechanical properties. The design of this hydrogel system is novel. The versatility in tuning the hydrogel's functionality using various cell-derived outer membrane vesicles and click chemistry combinations augments its potential applications.

Subsequent evaluations of the hydrogels' antibacterial and immunostimulatory functions have been presented. However, a notable observation is the predominant reliance on flow cytometry data in discussing most of the biological results. Instead of using Mean Fluorescence Intensity (MFI) values, % of expressions should be shown with representative histograms or dot plots. Furthermore, the observed significant enhancement in immunological activity when OMVs were incorporated into the hydrogel, as opposed to the activity of standalone OMVs, is an intriguing result. However, the discussions supporting the authors' biological claims appear overly simplified. A more exploration of these findings could enrich the manuscript and provide readers with deeper insights into the implications of these results.

Response: We thank the reviewer very much for her/his positive feedback on the novelty and significance of our work. We also appreciate these constructive suggestions from the reviewer regarding the analysis and discussion of the biological functionalities of the hydrogels. The raised concerns have been addressed appropriately, as detailed in the point-by-point response below.

- Figure 4 I and J: The figures depict the hydrogel before and after full stretching. However, the strain value isn't specified in the result section. If the hydrogel was stretched, one would expect a reduced number of OMVs. As presented, the figure might not accurately represent the change.

Response: We agree with the reviewer's point that stretching may lead to a reduced number of OMVs in the hydrogel. To validate our proposed hypothesis that the vesicles can deform to adapt to the stretching of the network (thus enhancing the mechanical toughness of the hydrogel), the hydrogel was stretched to approximately 400% strain and maintained in the stretched condition for freeze-drying. SEM was subsequently employed to investigate the changes in size, orientation, and morphology of OMVs under the stretched state. We have provided more details regarding the stretching strain in the Results section on page 11 and replaced Figure 4I and J with SEM images that better depict these alterations.

- Figure 5 C, D, E & Figure 5 I, J, K: A detailed discussion is needed to elucidate the factors that enhance the functionality of incorporated OMVs in the hydrogel system.

Response: We are grateful to the reviewer for her/his insightful comments. Regarding the superior antibacterial capabilities of SFSHs incorporated with OMVs than free OMVs, we speculated that this improvement might be attributed to the mechano-responsiveness of covalently crosslinked OMVs within the hydrogel. As hydrogel swelling could stretch SFSH network, the lipid bilayer of OMVs experienced disturbance, triggering a less tightly arranged phospholipid structure and subsequent the release of bioactive substances including antimicrobial proteins and toxins inherited from the parent cells, thereby endowing SFSHs with superior antibacterial ability. Similarly, the enhanced immunological activity of SFSHs might also be ascribed to the dynamic deformation of OMVs, which could induce the release of intrinsic antigens, inflammatory mediators, and other bioactive components from the OMVs. Detailed discussion has been supplemented on pages 12 and 14 in the revised manuscript.

- Flow Cytometry Data: Much of the biological data derives from flow cytometry. Histograms or dot plots should be included.

Response: We thank the reviewer for drawing our attention to this point. Regarding FCM data, in addition to MFI values, we have provided representative histograms or

dot plots. These have been highlighted in the revised version. Moreover, we have also supplemented the % of expressions in these FCM histograms and dot plots, including histograms depicting the surface modification of OMV-AM, OMV-N₃, and OMV_{SE}-N₃ (Figure 2A, 7A, and Figure S9A), as well as histograms and scatter plots illustrating the immunostimulatory functions of SFSHs (Figure S5-S7).

- Conclusion: If the intention is to emphasize the energy dissipation function of the hydrogel platform, it would be advantageous to perform cyclic tensile or compressive tests on hydrogels. Demonstrating that the loading curve does not overlap with the unloading curve would further validate energy dissipation claims.

Response: Basing on the reviewer's suggestion, the cyclic tensile tests were performed to verify the energy dissipation of SFSHs. Ten cycles of tensile loading and unloading at a strain of 1000% were subjected to SFSHs, while similar tests were conducted for the control polyacrylamide hydrogel at a strain of 400% due to the lower strain. Compared to the control hydrogel, the loading and unloading curves of the first cycle for SFSHs noticeably deviated with a significant elastic hysteresis loop, and the hysteresis circle and dissipated energy of the first cycle were markedly higher than those of the tenth cycle (Figure S3). These corroborated that the deformation and recovery of vesicular crosslinking points within SFSHs consumed a portion of energy, leading to non-coinciding tensile loading and unloading curves. The relevant details have been added in the revised manuscript (page 11).

Reviewer #3

Response: We thank the reviewer very much for taking his/her time to review our manuscript and help us to improve the quality of the work.

Reviewer #4

Wu and colleagues generated dual structurally and functionally skin-mimicking hydrogels by cross-linking cell-membrane compartments and demonstrated their elasticity, antibacterial, and immunological activity in vitro. This study is interesting but requires significant revisions.

Response: We thank the reviewer very much for her/his positive review of our work and providing many useful comments and constructive suggestions. The raised concerns have been addressed appropriately. Please see below for detailed point-by-point responses.

Specific Comments

1. "Why were only *Escherichia coli* Nissle 1917 cells used to generate outer membrane vesicles (OMVs)? This requires appropriate justification."

Response: Thanks for the reviewer's question. Actually, we have employed two types of extracellular vesicles from *Escherichia coli* Nissle 1917 and *Staphylococcus epidermidis* cells to construct SFSHs, and have regulated the structure and functionality of SFSHs by introducing different vesicles and crosslinking networks. Due to the easy availability, *Escherichia coli* Nissle 1917 cell-derived OMVs were employed as a model extracellular vesicle to fabricate SFSHs. The relevant details have been highlighted on page 6 in the revised manuscript.

2. Cells are removed by low-speed centrifugation, and in some cases several differential centrifugation steps are also performed to remove cell debris, large protein and membrane aggregates (<https://doi.org/10.1002/pmic.200700196> and <https://doi.org/10.1002/mas.20175>). Such steps were not followed by this study; the purity of OMVs is questionable, as they may not be free of cellular debris, large proteins, and membrane aggregates.

Response: We thank the reviewer for offering these insightful comments. As indicated in the references provided by the reviewer, a typical set of procedures including

differential centrifugation (9000 rpm, 20000 g, and 40000 g), sterile filtration through a 0.45 μm filter, ultracentrifugation (170000 g), and ultrafiltration (with a molecular weight cutoff of 100 kDa) is widely used for OMV purification. In fact, we have tried this purification method before. While, as shown in the below figure, the BCA protein quantification assay showed that the combination of differential centrifugation and ultrafiltration resulted in a similar protein content of the resulting solutions to the purification steps used in our work, but yielded a largely decreased number of OMVs. Given the fact that a large amount of OMVs was required to prepare the hydrogels, we chose the set of centrifugation (9000 rpm), sterile filtration through a 0.45 μm filter, and ultracentrifugation (170000 g) to collect OMVs.

3. In this study, the author used the ultracentrifugation method, which may contribute to non-OMV-associated proteins, unlike ultrafiltration techniques. In ultrafiltration, a bacterial supernatant is passed through a membrane with a given molecular weight cut-off, usually 50-100 kDa, which further questions the purity of OMVs.

Response: Yes, we agree with the reviewer's point that the use of ultrafiltration is able to remove non-OMV-associated proteins, while as answered to question 2, our method achieved a similar purification efficiency to the combination of ultracentrifugation and ultrafiltration, but with a much higher yield of OMVs. We speculated that the efficiency of OMV purification may also depend on bacterial strains that have different levels of extracellular proteins.

4. No purification of OMVs was performed after ultracentrifugation, such as Iodixanol (OptiPrep™) density gradient centrifugation or sucrose density gradient.

Response: Yes, Iodixanol density gradient centrifugation and sucrose density gradient are efficient approaches for OMV purification. We will definitely try these in our future works, particularly for the preparation of SFSHs for translation research. Thank you for your kind suggestions.

5. I would suggest that the authors include TEM images with multiple OMVs in figure 2D, or a single OMV that is appropriate to the average size of OMVs.

Response: According to the reviewer's suggestion, we have replaced Figure 2D with an image of a single OMV-AM with a similar size to the average size of OMVs.

6. Why didn't the author show TEM images of OMVs and OMV-AMs?

Response: Basing on the reviewer's comment, TEM images of OMVs and OMV-AM were separately shown in Figure S1A and Figure 2D.

7. I would suggest that the authors clarify the changes in the zeta potential of OMVs and OMV-AMs.

Response: The increase of zeta potential from -16.4 to -9.5 mV might be ascribed to the introduction of DSPE-PEG-AM, which inserted the hydrophobic DSPE tails into the lipid bilayer of OMVs and exposed the hydrophilic PEG-AM heads containing positively charged secondary amine groups on the surface of OMVs. This has been discussed in the revised manuscript (page 7).

8. Why was TEMED used to generate skin-mimicking hydrogels, even though it may be harmful upon contact with the skin?

Response: TEMED was utilized as a catalyst for initiating the radical polymerization reaction to construct SFSHs with polyacrylamide networks due to its high efficiency in initiating polymerization and compatibility with acrylamide chemistry. Actually,

TEMED was applied at an extreme low concentration of 5‰ w/w. As mentioned by the reviewer, if SFSHs were applied for in vivo applications, we can use additional procedures (e.g., diffusion) to remove TEMED. We thank the reviewer for drawing our attention to this point.

9. Why was only *Salmonella Typhimurium* used to assess the antibacterial ability of SFSHs

Response: This is an insightful comment. The rationalities are that Nissle 1917 cells have a defined antibacterial ability against *Salmonella Typhimurium* due to the secretion of microcin and OMV_{ECN} can inherit diverse bioactive substances such as microcin from their parent Nissle 1917 cells. We have clarified this in the revised manuscript (pages 11 and 12).

10. Importantly, this study lacks in vivo testing of their skin-mimicking hydrogels.

Response: We thank the reviewer for highlighting this and would like to emphasize that our work mainly focuses on the development of a cell-membrane compartment-crosslinking strategy to prepare dual structurally and functionally skin-mimicking hydrogels. We have systematically characterized the obtained hydrogels in the aspects of mechanical strength, antibacterial activity, and immune competence. The versatility of this strategy for expanding the structure and functionality of hydrogels has also been investigated by introducing a second cell-membrane compartment-crosslinked network. We agree with the reviewer that this type of hydrogels would show potential for in vivo applications and will definitely try in our future exploration as this is beyond the scope of the current work.

We thank all the reviewers again for taking their valuable time to review our manuscript. Their kind help and useful inputs are highly appreciated.

REVIEWERS' COMMENTS

Reviewer #1 (Remarks to the Author):

The reviewers' comments have been well addressed, I would like to recommend it to be accepted.

Reviewer #2 (Remarks to the Author):

The authors addressed all comments. One minor thing is adding gate strategies for the flow data in the supporting material.

Reviewer #4 (Remarks to the Author):

My concerns were well addressed by the authors.

Reviewer #1:

The reviewers' comments have been well addressed, I would like to recommend it to be accepted.

Reviewer #2:

The authors addressed all comments. One minor thing is adding gate strategies for the flow data in the supporting material.

Response: The gate strategies for the flow data have been added in Supplementary Figure 5 of the revised supporting information.

Reviewer #4:

My concerns were well addressed by the authors.

Response: We thank all the reviewers for their helpful and constructive comments.